# Higher Radiation Dose to the Immune Cells Correlates with Worse Tumor Control and Overall Survival in Patients with Stage III NSCLC: A Secondary Analysis of RTOG0617

**DOI:** 10.3390/cancers13246193

**Published:** 2021-12-08

**Authors:** Jian-Yue Jin, Chen Hu, Ying Xiao, Hong Zhang, Rebecca Paulus, Susannah G. Ellsworth, Steven E. Schild, Jeffrey A. Bogart, Michael Chris Dobelbower, Vivek S. Kavadi, Samir Narayan, Puneeth Iyengar, Cliff Robinson, Joel S. Greenberger, Christopher Koprowski, Mitchell Machtay, Walter Curran, Hak Choy, Jeffrey D. Bradley, Feng-Ming (Spring) Kong

**Affiliations:** 1Department of Radiation Oncology, Case Western Reserve University and University Hospitals of Cleveland, Cleveland, OH 44106, USA; Jian-Yue.Jin@UHhospitals.org; 2NRG Oncology Statistics and Data Management Center, Philadelphia, PA 19103, USA or huc@jhu.edu (C.H.); paulusr@nrgoncology.org (R.P.); 3Sidney Kimmel Comprehensive Cancer Center, Johns Hopkins University School of Medicine, Baltimore, MD 21218, USA; 4Abramson Cancer Center, University of Pennsylvania, Philadelphia, PA 19103, USA; Ying.Xiao@uphs.upenn.edu; 5Department of Radiation Oncology, School of Medicine, University of Maryland, Maryland, MD 20742, USA; Hong.Zhang@umm.edu; 6Department of Radiation Oncology, University of Pittsburgh Medical Center, Pittsburgh, PA 16251, USA; ellsworths3@upmc.edu (S.G.E.); greenbergerjs@upmc.edu (J.S.G.); 7Department of Radiation Oncology, Mayo Clinic Hospital, Phoenix, AZ 85054, USA; sschild@mayo.edu; 8Department of Radiation Oncology, State University of New York Upstate Medical University, Syracuse, NY 13210, USA; bogartj@upstate.edu; 9Department of Radiation Oncology, University of Alabama at Birmingham Cancer Center, Birmingham, AL 35233, USA; mdobelbower@uabmc.edu; 10USON-Texas Oncology-Sugar Land, Sugar Land, TX 77479, USA; vivek.kavadi@usoncology.com; 11Michigan Cancer Research Consortium CCOP, Ann Arbor, MI 48106, USA; Samir.Narayan@stjoeshealth.org; 12Department of Radiation Oncology, University of Texas Southwestern Medical School, Dallas, TX 75235, USA; Puneeth.Iyengar@UTSouthwestern.edu (P.I.); hak.choy@utsouthwestern.edu (H.C.); 13Department of Radiation Oncology, Washington University in St. Louis, St. Louis, MO 63108, USA; crobinson@radonc.wustl.edu (C.R.); jeffrey.d.bradley@emory.edu (J.D.B.); 14Christiana Care Health Services, Inc. CCOP, Newark, DE 19718, USA; ckoprowski@christianacare.org; 15Department of Radiation Oncology, Penn State University Cancer Institute, Hershey, PA 17033, USA; mmachtay@pennstatehealth.psu.edu; 16Department of Radiation Oncology, Winship Cancer Institute, Emory University, Atlanta, GA 30322, USA; wcurran@emory.edu; 17Department of Clinical Oncology, Hong Kong University Shenzhen Hospital, Shenzhen 518009, China; 18Department of Clinical Oncology, Queen Mary Hospital, Li Ka Shing Medical School, The University of Hong Kong, Hong Kong 999077, China

**Keywords:** non-small-cell lung cancer, radiotherapy, survival, radiation-induced immune toxicity

## Abstract

**Simple Summary:**

Emerging evidence indicates that the immune system plays an important role in controlling tumors during radiotherapy, and radiation-induced immune toxicity such as lymphopenia is associated with poor survival. However, the immune system is not considered as a critical organ at risk in radiotherapy partially because the radiation dose to the immune system is difficult to compute. In this study, we developed a model to compute the radiation dose to the circulating blood, which contains the majority of active immune cells. We then validated this model by examining the correlations of the blood dose with treatment outcome for patients enrolled in the NRG/RTOG0617 phase III clinical trial. We demonstrated that the blood dose was significantly and independently associated with overall survival and local progression-free survival. This result suggests that radiation dose to circulating immune cells is critical for tumor control, and decreasing the dose to the immune system has the potential to improve survival.

**Abstract:**

**Background**: We hypothesized that the Effective radiation Dose to the Immune Cells (EDIC) in circulating blood is a significant factor for the treatment outcome in patients with locally advanced non-small-cell lung cancer (NSCLC). **Methods**: This is a secondary study of a phase III trial, NRG/RTOG 0617, in patients with stage III NSCLC treated with radiation-based treatment. The EDIC was computed as equivalent uniform dose to the entire blood based on radiation doses to all blood-containing organs, with consideration of blood flow and fractionation effect. The primary endpoint was overall survival (OS), and the secondary endpoints were progression-free survival (PFS) and local progression-free survival (LPFS). The EDIC–survival relationship was analyzed with consideration of clinical significant factors. **Results**: A total of 456 patients were eligible. The median EDIC values were 5.6 Gy (range, 2.1–12.2 Gy) and 6.3 Gy (2.1–11.6 Gy) for the low- and high-dose groups, respectively. The EDIC was significantly associated with OS (hazard ratio [HR] = 1.12, *p* = 0.005) and LPFS (HR = 1.09, *p* = 0.02) but PFS (HR = 1.05, *p* = 0.17) after adjustment for tumor dose, gross tumor volume and other factors. OS decreased with an increasing EDIC in a non-linear pattern: the two-year OS decreased first with a slope of 8%/Gy when the EDIC < 6 Gy, remained relatively unchanged when the EDIC was 6–8 Gy, and followed by a further reduction with a slope of 12%/Gy when the EDIC > 8 Gy. **Conclusions**: The EDIC is a significant independent risk factor for poor OS and LPFS in RTOG 0617 patients with stage III NSCLC, suggesting that radiation dose to circulating immune cells is critical for tumor control. Organ at risk for the immune system should be considered during RT plan.

## 1. Introduction

Lung cancer is the leading cause of cancer-related death. Over 85% of lung cancer cases are non-small-cell lung cancer (NSCLC) [1], 40% of which are stage III [1,2]. The standard of care for unresectable stage III NSCLC is radiotherapy (RT) and concurrent chemotherapy [3], with recent data supporting the routine integration of adjuvant immunotherapy [4]. Despite advances in RT technology, treatment outcomes remain suboptimal, and local disease progression is a major cause of death [4]. Intensifying local therapy with RT dose escalation was therefore proposed to improve local tumor control and survival [5]. However, RTOG 0617, the largest study to date on dose–response effects in stage III NSCLC, demonstrated significantly worse overall survival (OS) in patients treated on the high-dose arm (74 Gy) vs. the low-dose arm (60 Gy) [6].

Radiation is known for immune modulation effect. Radiation-induced tumor cell killing can activate the immune system through various mechanisms including releasing tumor-specific antigens [7]. Preclinical studies have demonstrated that the immune system plays a key role in tumor control after RT [8,9]. Treatments with RT alone or RT combined with immunotherapy can control tumors in immunocompetent mice, but not in immune-deficient ones [8]. An abscopal effect (i.e., shrinkage of un-irradiated tumors outside the RT field) has been observed in animal studies [8,9] and in clinical settings [10]. While these observations suggest that RT may augment anti-tumor immunity in special situations, RT is also well known for its immunosuppressive effects. One of the most common and clinically significant features of radiation-induced immunosuppression is radiation-induced lymphopenia [11,12], which has been repeatedly reported as a risk factor for poorer survival in several cancers, and a recent pooled analysis reported significance in patients with multiple treatment-refractory solid tumors including NSCLC [11,12,13,14,15].

However, the immune system has not yet been considered as a critical organ at risk for RT planning in practice, though RT dose escalation to tumor is also expected to increase the radiation dose to immune-related structures. We hypothesized that high radiation dose to the immune system may impair various immune functions including anti-tumor immunity, consequently decreasing tumor control. Specifically, we hypothesized that circulating immune cells in blood are an important target for radiation-induced immunosuppression because: (1) radiation-induced lymphopenia occurs following breast and brain radiation, where the treated field contains little marrow or lymphatic tissue [16,17,18]; (2) circulating lymphocytes in blood are one of the most radiosensitive cell types. To test these hypotheses, we first developed a model to compute Effective Dose to Immune Cells in circulating blood (EDIC) as an estimation of equivalent uniform dose to the entire blood during the RT course, and then evaluated the relationship between the EDIC and the risks of local tumor progression and death. The RTOG 0617 trial provided an ideal setting for testing this hypothesis, as detailed dosimetric and survival data were available for nearly all patients enrolled in this large phase III cooperative group trial treated with high and low radiation doses.

## 2. Materials and Methods

### 2.1. Clinical and Dosimetric Patient Data

RTOG 0617 was a phase III trial for unresectable stage III NSCLC. The study details and primary analysis results have been published previously [6]. Briefly, all patients received conformal radiotherapy with concurrent and consolidation chemotherapy (carboplatin and paclitaxel). A two-by-two factorial randomized design was used to assign patients to two different RT dose groups (60 Gy vs. 74 Gy) and systemic therapy arms (carboplatin/paclitaxel with or without cetuximab). Patients included in this secondary analysis must have retrievable RT plans and must have received a confirmed dose of at least 50 Gy, which was an arbitrary cut-off chosen to identify patients who had received a clinically meaningful RT dose. Clinical factors tested for confounding effects included baseline Zubrod performance status, use of positron emission tomography (PET) during staging, tumor histology, age at randomization, gender, race, tumor location(s), weight loss, smoking history, gross tumor volume (GTV), and whether the full course of chemotherapy was received. Conventional radiation dosimetric data such as mean lung dose (MLD), mean heart dose (MHD) and integral total dose volume (ITDV) were included for modeling. ITDV is the integration of dose in the total irradiated volume in the scanned CT image.

### 2.2. EDIC Computation

The immune system is a complex interplay of multiple features and functions of immune cells. Lymphocytes are the most important functional immune cells. Lymphocytes originate in the bone marrow and/or thymus and circulate through the body into various organs via blood vessels, and may return to the blood circulation through lymph nodes and lymphatic ducts. They also circulate through other lymphatic organs such as the spleen. In addition, lymphocytes (specifically T cells) may reside within the tissue of various organs [19]. They even reside within tumor tissue as infiltrated T cells and play a key role in RT-mediated anti-tumor immunity [19]. In this study, we considered the lymphocytes in circulating blood as a key target for radiation-induced suppression in the anti-tumor immunity because the tumor-resident T cells are very radio-resistant [19], and the lymphocytes in circulating blood provide continuous supply for the loss of tumor-resident T cells during their anti-tumor activities. However, it is very challenging to determine the radiation dose to the lymphocytes in circulating blood because they are moving targets. Using a blood-flow continuity principle, we first calculated the blood dose and volume contributed by a single fraction of radiation to each particular blood-containing organ, including lung, heart, great vessels and body mass. The fractionation effect of irradiation on flowing blood through each organ was modelled using a similar approach reported by Yovino et al. [18]. The blood dose volume was then converted into an equivalent uniform dose (EUD). The total effective blood dose was the sum of the EDUs from the contributions of all irradiated organs. We defined this sum of EUDs as the EDIC, the Effective Dose to the Immune Cells in circulating blood. This EDIC model was presented in the 2017 ASTRO annual meeting [20]. A detailed description of the approach to EDIC derivation is described in the Appendix B. The EDIC is finally expressed as the following equation for patients receiving ≥25 fractions of thoracic radiation:(1)$$EDIC=B_{1}\%*MLD+B_{2}\%*MHD+\left[B_{3}\%+B_{4}\%*k_{1}*{\left(\frac{n}{45}\right)}^{\frac{1}{2}}\right]*ITDV/\left(61.8*10^{3}\right)$$
where *B*_1_% = 0.12, *B*_2_% = 0.08, *B*_3_% = 0.45 and *B*_4_% = 0.35 represent the percentages of blood volume within the four major blood-containing organs (lung, heart, great vessels, and small vessels/capillaries in all other organs, respectively) out of the total blood volume in the body; MLD, MHD and ITDV are the mean lung dose, mean heart dose and integral total dose volume; *k*_1_ = 0.85 is a dose effectiveness factor due to the small percentage of cardiac output for the small vessels/capillaries; and 61.8 × 10^3^ (cm^3^) is the average total body volume, assuming average weight of 63 kg (140 lbs) and density of 1.02 g/cm^3^.

### 2.3. Outcomes and Statistical Considerations

The primary endpoint was overall survival (OS). The progression-free survival (PFS) and local progression-free survival (LPFS) were tested as secondary endpoints. These endpoints were analyzed as time-to-event data and calculated from the date of randomization to the date of respective event or last follow-up. The OS event was death due to any cause; the PFS event was the first occurrence of any progression or death; and the LPFS event was the first occurrence of local failure or death. These rates were estimated using the Kaplan–Meier method, and the distributions between different groups were compared using the log-rank test. Cox proportional hazards models were used to evaluate the relationships between the EDIC and other factors with OS, PFS and LPFS. Because the EDIC was derived from the combination of MLD, MHD and ITDV, these variables were evaluated individually under multivariable analyses to avoid potential collinearity. The tumor dose effect was adjusted by stratification in both univariate and multivariate analyses. The functional forms of the EDIC in the Cox models were explored both linearly and using restricted cubic splines [21]. To illustrate the non-linear functional form of the EDIC in the Cox model, the EDIC was also categorized based on quartiles and absolute EDIC values. The proportionality assumption was graphically assessed using plots of log(-log[survival]) versus log of survival time, and tested using a formal test based on the Schoenfeld residuals [22]. Interaction terms (e.g., potentially differential effects of the EDIC on outcomes by different levels of patient characteristics) were also examined using the Wald test.

All statistical tests were two-sided and performed using SAS 9.4 software (SAS Institute, Cary, NC, USA) and R version 3.4.0 (available at https://www.r-project.org/ (accessed on 11 July 2017). *p* < 0.05 was considered statistically significant.

## 3. Results

### 3.1. Patient Characteristics and the EDIC

Of 495 patients enrolled in RTOG 0617, 29 were excluded due to failure of RT plan retrieval. Ten additional patients were excluded due to errors in plan archiving or other reasons: RT plans missed the target (*n* = 1), wrong RT plans were archived (exact same plans for 2 different patients; *n* = 4), or the total dose received was ≤50 Gy (*n* = 5). Of the 456 eligible patients, 256 and 200 were originally assigned to the standard (60 Gy) and high-dose arms (74 Gy), respectively; 261 patients received 60 Gy (including some originally assigned to the high-dose arm), 165 received 74 Gy, 4 received 52–58 Gy, 12 received 62–66 Gy, and 6 received 67–72 Gy. Patients were categorized according to the actual dose received, with a high dose defined as ≥67 Gy. Based on this definition, 285 and 171 patients were placed into the low- and high-dose groups, respectively.

The median follow-up time for patients alive at the last evaluation was 30.3 months (range, 2.5–61.5 months). Demographic, clinical, and dosimetric data for the included patients are summarized in Appendix A.

The EDIC was calculated for all 456 patients. The median EDIC values were 5.6 Gy (range, 2.1–12.2 Gy) for the low-dose group, 6.3 Gy (2.1–11.6 Gy) for the high-dose group, and 5.9 Gy (2.1–12.2 Gy) for all patients.

### 3.2. Univariate Analysis of the EDIC and Clinical Factors for OS, PFS and LPFS

EDIC was significantly associated with OS, PFS and LPFS (Table 1) after adjusting for the tumor dose effect using stratification. Similar to previously reported results, gender, Zubrod performance status, tumor histology, smoking history, use of PET staging, and American Joint Committee on Cancer (AJCC) stage were not significantly associated with OS, PFS and LPFS, whereas the occurrence of grade ≥ 3 esophagitis/dysphagia and completion of the full course of chemotherapy were significantly associated with OS, PFS and LPFS (Table 1). According to the dose that the patients actually received, the low-dose patients had marginally better OS (*p* = 0.10) and PFS (*p* = 0.08), and significantly better LPFS (*p* = 0.02) than the high-dose patients. Tumor location (central/lower left lobe vs. others) was significantly associated with OS only. Interestingly, all dosimetric factors, including GTV, MLD, and MHD, were significantly associated with OS, PFS and LPFS; ITDV was significant for OS but PFS and LPFS (Table 1).

### 3.3. The EDIC in Multivariable Analysis of Factors Associated with OS, PFS and LPFS

The EDIC effects were assessed under two different multivariable models: one without the EDIC but with MLD/MHD/ITDV and one with the EDIC but without MLD/MHD/ITDV, as there were apparent correlations between MLD/MHD/ITDV and the EDIC. Other clinical and dosimetric factors, including tumor location, gross tumor volume, esophagitis grade, and received full chemo, were considered for these multivariable analyses. The tumor dose effect was adjusted by stratification according to the actual received radiation dose. The occurrence of grade ≥ 3 esophagitis/dysphagia and completion of chemotherapy were significant for OS, PFS and LPFS in both models (Table 2a–c). Interestingly, in the multivariable Cox model of OS without the EDIC, only GTV was significant but not MHD, MHD and ITDV. On the other hand, in the multivariable Cox model of OS with the EDIC, the EDIC was significant, while GTV was not (Table 2a).

However, the multivariable PFS model was quite different from the OS model (Table 2b). Both MLD and GTV were significantly associated with PFS in the model without the EDIC, while both GTV and the EDIC were not significant in the model with the EDIC. The multivariable LPFS model was similar to the OS model (Table 2c). The EDIC was a significant factor for LPFS while GTV was not in the model with the EDIC. However, MHD was a significant factor while MLD and GTV were not in the LPFS model without the EDIC.

### 3.4. Non-Linear Relationship between the EDIC and OS

To further analyze the relationship between the EDIC and OS, 456 patients were divided into four groups with equal number of patients per the quartiles of the EDIC (Figure 1a), as well as into six groups with an equal dose increment of 1.5 Gy between two groups (Figure 1b). The Kaplan–Meier curves shown in Figure 1 depict a strong inverse relationship between OS and the EDIC (i.e., the greater the EDIC, the worse the OS). However, the relationship between the EDIC and OS was not strictly linear, because the survival estimates of some intermediate EDIC groups overlap in both Figure 1a,b. To further illustrate this non-linear relationship, we estimated the hazard of the EDIC by nonparametric smoothing using restricted cubic splines in a univariate Cox regression model of OS, stratified tumor dose (Figure 2a). This analysis showed that hazard rates increased with increasing EDIC when the EDIC was less than 6.0 Gy or larger than 8.0 Gy but not between 6 and 8 Gy. This Cox regression analysis showed that the risk of death increased by 23%/Gy (HR = 1.23, 95% CI:1.07–1.41; *p* = 0.003) with increasing EDIC when the EDIC < 6.0 and by 37%/Gy (HR = 1.37, 95% CI: 1.14–1.64; *p* = 0.0007) when the EDIC > 8 Gy. However, this curve was relatively flat for EDIC values in the range of 6.0–8.0 Gy.

This non-linear relationship was also illustrated by the survival dose–response curve for 2 year OS versus the EDIC (Figure 2b). The six data points were determined from the data of the six subgroups in Figure 1b, with the horizontal axis of each point being the average EDIC for the corresponding subgroup. The data were well fitted by an OS model composed of two normal tissue complication probability (NTCP) components:(2)$$OS=0.74*\left[1-\frac{0.39}{1+{\left(\frac{4.5}{EDIC}\right)}^{6}}\right]*\left[1-\frac{1}{1+{\left(\frac{9.9}{EDIC}\right)}^{12}}\right]$$
with D_50_ being 4.5 and 9.9 Gy, respectively, for the two NTCP components. Alternatively, the survival dose–response can also be described by a combined linear model in three parts: (1) for an EDIC < 6.0 Gy, 2 year OS decreases with increasing EDIC at a slope of 8%/Gy; (2) for an EDIC of 6.0–8.0 Gy, OS does not vary; and (3) for an EDIC > 8.0 Gy, 2 year OS decreases with increasing EDIC at a slope of 12%/Gy.

## 4. Discussion

In this study of a secondary analysis of a multicenter phase III study, we presented a novel approach for approximating the effective radiation dose to immune cells in circulating blood (abbreviated EDIC). Our results demonstrated an overall inverse relationship between the EDIC and survival outcome in OS and LPFS in a large cohort of patients enrolled in RTOG 617, which is the largest study to date of RT dose response in stage III NSCLC. Multivariable analysis demonstrated that this correlation remained significant after adjustment for other known prognostic factors including tumor dose and GTV, while heart and lung doses were no longer significantly associated with OS after adjusting for these factors. These results support our hypothesis that a high radiation dose to the host immune system, particularly to immune cells in the circulating blood, may impair anti-tumor immunity, worsening tumor control and survival. This study appears to provide a reasonable explanation for the unexpected worse OS and LPFS from higher dose in patients enrolled the RTOG 0617 trial, suggesting that radiation-induced immunosuppression, rather than radiation-induced grade-5 cardiac/pulmonary toxicity, may account for the poorer OS in the high-dose arm. It should also be noted that only a few grade-5 toxicity cases were reported in the RTOG 0617 trial [6].

High lung and heart RT doses were reported to have significant correlations with decreased OS [23,24], which seemed to support the assumption that lung and heart toxicity had contributed to the worse survival seen in the high-dose arm of RTOG 0617. However, our results on actually received doses showed that high MLD and MHD were significantly correlated with OS only under univariate analyses, not in the multivariable Cox model. MLD was significantly correlated with only PFS, while MHD was significant for LPFS under the same multivariable models, suggesting that MLD or MHD is associated with survival due to disease control/progression rather than toxicity. Furthermore, in RTOG 0617, the high-dose arm had less pulmonary toxicity than the low-dose arm, the total reported grade 3+ pulmonary toxicity was less than 5%, and there were only a few pulmonary deaths and no reported cardiac deaths [6]. Therefore, the potential correlation of lung and heart doses to survival is likely not directly due to conventional heart and lung toxicity. Rather, they may be surrogates of radiation dose to immune structures, such as circulating immune cells in blood and other components of the immune system, which are critical for tumor control. For example, MLD could also be a surrogate of doses to resident lymphocytes in lung tissue and to pulmonary lymph nodes. The significance of this study is the development of the EDIC model, and the validation of its independent correlation with OS in 456 patients with NSCLC. This EDIC model (first presented in ASTRO 2017) was also adopted and validated externally in 117 patients with non-small-cell lung cancer [25], 92 patients with esophageal cancer treated with neoadjuvant chemoradiation [26] and 488 patients with esophageal cancer treated with concurrent chemoradiation [27].

One may also ask if the EDIC effect could be a result of tumor volume, which is a potential confounder, since larger tumors are considered to be associated with worse survival. However, our multivariable analysis accounts for this potential confounder by showing that the EDIC was still significantly associated with OS and LPFS after adjusting for GTV size effects. Furthermore, GTV was no longer a significant adverse prognostic factor when it was analyzed together with the EDIC. This result suggests that tumor volume is not likely to explain our observations. It is understandable that the EDIC is a combined effect of tumor volume and “goodness” of the RT plan, i.e., patients with relative small tumors may have a greater EDIC if the plan was not optimized or the tumor was in a special location. The greater significance of the EDIC over GTV suggests the importance of RT planning. Additionally, our data also showed a significant effect of grade ≥ 3 esophagitis on OS, PFS and LPFS after adjusting for GTV size effects. Esophagitis would certainly not cause tumor progression. However, severe esophagitis may indicate that the patient is more radiosensitive, and a more radiosensitive patient may have a greater risk of severe radiation-induced immunosuppression and thus an increased risk of tumor progression and poorer survival. Completion of the entire chemotherapy course was also a significant factor for OS, PFS and LPFS. Of the total 456 patients, 58 patients did not complete the chemotherapy course. These patients might be not able to tolerate the treatment. It is understandable that they had a worse treatment outcome.

The key clinical implication of this study is that circulating immune cells may be considered as a critical organ at risk during external-beam RT, and that the EDIC is a potentially useful parameter for plan optimization to limit incidental toxicity to circulating immune cells during RT. Several approaches have the potential to reduce the EDIC, including the following: (1) reduction in circulating blood exposure via hypofractionated treatment regimens and/or decrease in the radiation delivery time, (e.g., with very high-dose-rate techniques such as FLASH RT) [28]; (2) optimization of plans by adjustment of beam energies and directions, number of beams, and collimator margins, as well as the use of intensity-modulated radiotherapy (IMRT) and other advanced planning techniques; (3) the use of advanced RT technology, such as image-guided adaptive therapy [29] and proton therapy [30]; and (4) dose de-escalation and/or margin reduction, tailored to individual radiosensitivity [31].

The present study has limitations. Firstly, it did not have blood count data to show correlation of the EDIC with the lymphocyte count. However, the lymphocyte count may not well reflect the radiation killing of immune cells in blood as well as the potential anti-tumor immunity, because lymphocytes can be replenished from the spleen and bone marrow, and there are many subpopulations, such as B cells, T cells, CD8, CD4, CD4 helper, CD4 regulator, NK cells, memory cells, as well as proliferating and non-proliferating cells for each subset in the lymphocytes. These cells have a large variation of radiosensitivity [32]. Russ et al. showed that a very low-dose (0.01~0.1 Gy) total body irradiation (TBI) in rats reduced the lymphocyte count to 50% of the baseline level in hours, and the lymphocyte count recovered to the baseline level several days later [33]. They also showed that multiple low-dose TBIs with weeks interval increased lymphocyte count several times higher than the baseline level, and these rats accepted syngeneic tumor implants (poor immunity), while the rats with lymphocytes recovery to the baseline level rejected syngeneic tumor implants (good immunity). These data support that the lymphocyte count after radiation does not necessary correlate with the radiation dose, and a higher lymphocyte count does not mean a better anti-tumor immunity. In addition, So et al. and Xu et al. have used the EDIC model to study its association with lymphopenia and OS in esophageal cancers [26,27]. Both studies found a significant association of the EDIC with lymphopenia. More importantly, they found that the EDIC was much more predictive of OS and distant metastasis-free survival than the lymphopenia [27].

Secondly, the EDIC model was based on relatively simple assumptions, including the following: (1) integral dose was used to approximate the mean dose to the large vessels and small vessels/capillaries; (2) an approximate model was used to calculate the EUD contribution to small vessels and capillaries in other organs; (3) the percentages of blood volume assigned to lung, heart, great vessels, and small vessels/capillaries may not reflect the actual blood distribution in the body; (4) average rather than patient-specific parameters were used for approximating the dose contributions; (5) the ITDV did not exclude the lung and heart volumes. These assumptions would affect the accuracy of the model. For example, the great vessels contain a high concentration of lymphocytes. However, our study was initially a proof of concept study. In addition, we expect that more precise values for these parameters would not significantly improve the accuracy of the EDIC model, because the EDIC model is relatively accurate and reliable for its main contributors: the lung and heart. While the EDIC is the estimation of the physical dose to the circulating immune cells in blood, it can also be considered as a parameter that optimally combines three independent factors, the MLD, MHD and ITDV. Additionally, it is notable that this study did not account for the contribution of the other immune substructures such as lymph nodes, lymphatic ducts (particularly the thoracic duct), thymus, bone marrow, and resident lymphocytes in lung and tumor tissue. Future work shall develop and validate a more comprehensive model, including carefully determination of doses to large vessels, lymphatic organs, and resident immune cells in lung and tumor.

## 5. Conclusions

This study demonstrated that the EDIC is significantly associated with local progression-free survival and overall survival in a large prospective cooperative group study of concurrent chemoradiation for the treatment of unresectable stage III NSCLC. These findings suggest that radiation dose to the immune cells is a critical determinant of treatment outcomes. While an external validation study is needed, the knowledge gained from the present study may be used to guide RT planning to improve tumor control and survival in patients with locally advanced NSCLC.

## Figures and Tables

**Figure 1 cancers-13-06193-f001:**
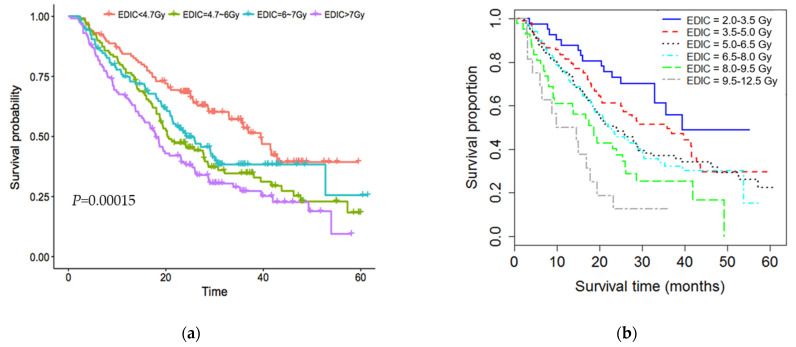
Overall survival curves separated by the effective dose to immune cells (EDIC). (**a**) for patients divided into 4 quartiles according to the EDIC); and (**b**) for patients divided into 6 EDIC groups with a 1.5 Gy dose increment. Survival improved significantly with a reduction in the EDIC. However, overall survival (OS) were not significantly different among patients with an EDIC between 6.0 and 8.0 Gy.

**Figure 2 cancers-13-06193-f002:**
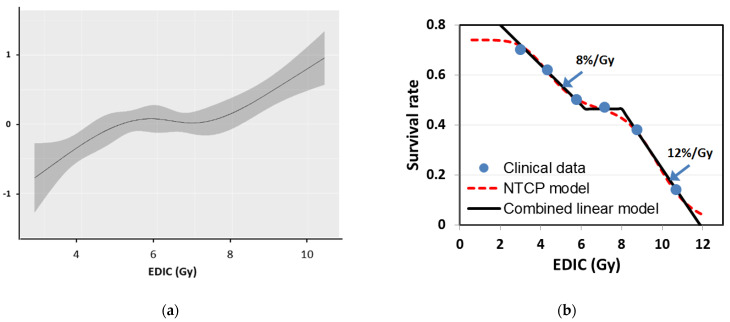
Quantitative effect of the effective dose to immune cells (EDIC) on the risk of death. (**a**) Relationship between relative hazard of death and the EDIC. The hazard of death increased with increasing EDIC when EDIC values was <5.5 Gy and remained relatively unchanged when EDIC values ranged 5.5 to 7.5 Gy, and again increased with EDIC values > 7.5 Gy. (**b**) Relationship between 2 year overall survival (OS) rate and the EDIC by a normal tissue complication probability (NTCP) survival model. The clinical data were well fitted by the NTCP model composed of two components, with D50 being 4.5 and 9.9 Gy, respectively. The clinical data could also be described by a combined linear model with 3 parts: (1) for an EDIC < 6.0 Gy, 2 year OS decreased with increasing EDIC at a slope of 8%/Gy; (2) for an EDIC between 6.0 and 8.0 Gy, 2 year OS remained unchanged; and (3) for an EDIC > 8.0 Gy, 2 year OS decreased with increasing EDIC at a slope of 12%/Gy.

**Figure A1 cancers-13-06193-f0A1:**
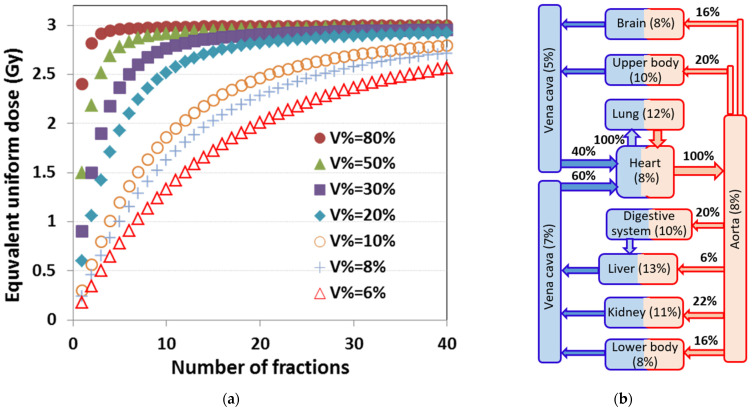
Computation of the EDIC model. (**a**) Equivalent uniform dose (EUD) to the total blood varies with fraction number (*n*) for various *V*% (the percentage blood volume irradiated in a specific organ during each fraction), which contains the *B*% of blood volume that receives irradiation with a mean organ dose (MOD). We have *MOD* ∗ *B*% = 3 Gy for this example organ. *V*% depends on the cardiac output *A*%, blood volume *B*%, blood circulation time *T*, and beam-on time *t* for each fraction and can be determined as *V*% = *B*% + (*A*% − *B*%) ∗ *t*/*T*. We note that when *n* > 20 and *V*% > 20%, *EUD*~*MOD* ∗ *B*%. Generally, EUD can be approximated as *EUD*~*MOD* ∗ *B*% ∗ *k*_1_ ∗ (*n*/45)^1/2^ for other organs with a smaller *V*%, where *k*_1_ is a dose effectiveness factor. (**b**) Major organs in the blood circulation system and their estimated percentage cardiac output (*A*%) and percentage blood volume (*B*%) based on an anatomy/physiology text book. For thoracic irradiation, 4 organs/components, including lung, heart, great vessels, and small vessels/capillaries in other organs, contribute to the total blood dose. Because *A*% ≥ 40% for lung, heart and great vessels, and with *t*~1 min and *T*~1 min, we have *V*% = *A*% ≥ 40%, and the contribution of dose to the blood for these organs can be estimated to be *EUD*~*MOD* ∗ *B*%.

**Table 1 cancers-13-06193-t001:** Univariate analysis with stratification of patients by actual received RT dose *.

Variables	OS		PFS		LPFS	
	HR (95% CI)	*p*	HR (95% CI)	*p*	HR (95% CI)	*p*
Prescription dose: cont.	1.31 (1.04, 1.67)	0.01	1.22 (0.98, 1.51)	0.07	1.34 (1.07–1.67)	0.01
Actual received dose: cont.	1.22 (0.95, 1.56)	0.10	1.21 (0.98, 1.50)	0.08	1.32 (1.05, 1.65)	0.017
Age: cont.	1.01 (0.99, 1.02)	0.24	0.997 (0.99, 1.009)	0.59	1.01 (0.99, 1.02)	0.26
Gender: Male (RL) vs. Female	0.83 (0.65, 1.06)	0.13	0.96 (0.77, 1.19)	0.71	0.88 (0.70, 1.10)	0.27
Zubrod status: 0 (RL) vs. 1	1.02 (0.80, 1.30)	0.86	0.95 (0.76, 1.18)	0.64	1.01 (0.81, 1.27)	0.91
Histology: Non-Sq (RL) vs. Sq	1.13 (0.88, 1.43)	0.34	1.02 (0.82, 1.26)	0.87	1.19 (0.95, 1.49)	0.12
Smoking history: Yes (RL) vs. No	0.72 (0.43, 1.21)	0.22	0.79 (0.50, 1.26)	0.32	0.80 (0.50, 1.30)	0.37
RT technique: 3D (RL) vs. IMRT	0.89 (0.70, 1.13)	0.33	1.04 (0.84, 1.28)	0.74	1.06 (0.84, 1.32)	0.64
PET staging: No (RL) vs. Yes	0.76 (0.52, 1.11)	0.16	0.87 (0.61, 1.24)	0.45	0.83 (0.58, 1.21)	0.34
AJCC stage: IIIA (RL) vs. IIIB	1.03 (0.80, 1.32)	0.82	1.08 (0.86, 1.35)	0.52	1.08 (0.86, 1.37)	0.49
Tumor location: Not LLL/central (RL) vs. LLL/central	1.49 (1.06, 2.09)	0.02	1.21 (0.88, 1.66)	0.25	1.33 (0.95, 1.84)	0.09
Esophagitis grade: <3 (RL) vs. ≥3	1.77 (1.30, 2.41)	0.0003	1.72 (1.29, 2.28)	0.0002	1.53 (1/14. 2.06)	0.005
Received full chemo: No (RL) vs. Yes	0.64 (0.46, 0.90)	0.009	0.72 (0.53, 0.97)	0.03	0.70 (0.51, 0.97)	0.03
GTV: cont.	1.21 (1.07, 1.38)	0.0026	1.13 (1.01, 1.26)	0.03	1.13 (1.01, 1.27)	0.04
Mean lung dose: cont.	1.05 (1.02, 1.09)	0.0004	1.04 (1.01, 1.07)	0.003	1.03 (1.004, 1.06)	0.02
Mean heart dose: cont.	1.02 (1.01, 1.03)	<0.0001	1.01 (1.003, 1.02)	0.004	1.02 (1.007, 1.03)	0.0007
Integral total dose: cont.	1.003 (1.001, 1.005)	0.0004	1.001 (1.00, 1.003)	0.11	1.002 (1.00, 1.003)	0.03
EDIC: cont.	1.18 (1.10, 1.26)	<0.0001	1.10 (1.03, 1.16)	0.002	1.11 (1.05, 1.18)	0.0009

* The effect of actual received dose has been stratified for all other factors in this univariate analysis except for the prescription dose and actual received dose. Abbreviations: cont., continuous variable; RL, reference level; Sq, squamous; LLL, left lower lobe; RT, radiotherapy; OS, overall survival; PFS, progression-free survival; LPFS, local progression-free survival; HR, hazard ratio; CI, confidence interval; GTV, gross tumor volume; EDIC, Effective Dose to Immune Cells.

**Table 2 cancers-13-06193-t002:** Stratified multivariable analyses with stratification of patients according to the actual received RT dose.

(a)				
Variables	OS without EDIC		OS with EDIC	
	HR (95% CI)	*p*	HR (95% CI)	*p*
Tumor location	1.42 (0.98, 2.05)	0.07	1.41 (0.98, 2.02)	0.07
Gross tumor volume	1.16 (1.00, 1.34)	0.05	1.12 (0.98, 1.28)	0.09
Esophagitis grade	1.53 (1.11, 2.11)	0.01	1.52 (1.10, 2.10)	0.012
Received full chemo	0.58 (0.41, 0.81)	0.0015	0.59 (0.42, 0.83)	0.003
Mean lung dose	1.03 (0.998, 1.070)	0.07		
Mean heart dose	1.008 (0.995, 1.022)	0.21		
Integral total dose	1.000 (0.998, 1.002)	0.93		
EDIC			1.12 (1.03, 1.21)	0.005
**(b)**				
**Variables**	**PFS without EDIC**		**PFS with EDIC**	
	**HR (95% CI)**	** *p* **	**HR (95% CI)**	** *p* **
Tumor location	1.19 (0.84, 1.68)	0.33	1.20 (0.85, 1.68)	0.30
Gross tumor volume	1.15 (1.01, 1.32)	0.04	1.08 (0.96, 1.21)	0.20
Esophagitis grade	1.64 (1.22, 2.21)	0.001	1.60 (1.19, 2.15)	0.002
Received full chemo	0.63 (0.46, 0.86)	0.003	0.66 (0.49, 0.90)	0.009
Mean lung dose	1.04 (1.006, 1.071)	0.02		
Mean heart dose	1.005 (0.992, 1.017)	0.47		
Integral total dose	0.998 (0.996, 1.000)	0.10		
EDIC			1.05 (0.98, 1.12)	0.17
**(c)**				
**Variables**	**LPFS without EDIC**		**LPFS with EDIC**	
	**HR (95% CI)**	** *p* **	**HR (95% CI)**	** *p* **
Gross tumor volume	1.10 (0.96, 1.26)	0.16	1.07 (0.95, 1.20)	0.29
Esophagitis grade	1.36 (1.00, 1.85)	0.05	1.37 (1.00, 1.86)	0.05
Received full chemo	0.66 (0.48, 0.91)	0.012	0.67 (0.48, 0.92)	0.013
Mean lung dose	1.01 (0.98, 1.04)	0.48		
Mean heart dose	1.012 (1.000, 1.024)	0.044		
Integral total dose	1.00 (0.998, 1.002)	0.81		
EDIC			1.09 (1.01, 1.16)	0.02

Abbreviations: OS, overall survival; HR, hazard ratio; CI, confidence interval; EDIC, effective dose to immune cells; PFS, progression-free survival; LPFS, local progression-free survival.

## Data Availability

The data presented in this study are available on reasonable request from the corresponding author and NRG Oncology.

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
