# Peer review of "Higher Radiation Dose to the Immune Cells Correlates with Worse Tumor Control and Overall Survival in Patients with Stage III NSCLC: A Secondary Analysis of RTOG0617"

_cancers, 2021, doi:10.3390/cancers13246193_

Round 1

Reviewer 1 Report

This is an interesting manuscript examining correlations between the approximated dose received by circulating blood (and therefore by extension, circulating immune cells) and oncologic outcomes in RTOG 0617. The authors find in this post-hoc analysis that higher EDIC is negatively correlated with overall survival and local PFS. Overall the methodology and results are presented well. Some comments:

  1. In the calculation of EDIC, why is a standardized patient weight (63 kg) used, instead of the actual weights for each patient? It seems this information should be available.
  2. When incorporating the integral total dose volume (ITDV) in the EDIC calculation, does ITDV exclude the dose received by heart and lungs? The text seems to define ITDV as just the integrated dose across the total irradiated volume, which would include the doses received by heart and lungs - in which case, it seems that those doses are being counted twice in the estimation of EDIC.
  3. Without also being able to correlate EDIC to lymphocyte count (or other immune cell endpoint), would temper the overall conclusions regarding radiation dose to immune cells being the explanatory factor for the results of RTOG 0617. In particular, other work has shown that heart dose is linked to early cardiac morbidity in lung cancer patients (even despite the competing risk of lung cancer death), so even though heart dose fell out of the multivariable analyses here, I think showing a correlation with lymphocyte count would really strengthen that argument that EDIC is truly an explanatory factor for the results of 0617. 
  4. Do the authors have a biological rationale for why overall survival would plateau between 6-8 Gy EDIC? It doesn't make much sense to me why this should be the case, and I wonder if it's more the result of overfitting the data than a real effect.
  5. Between Table 1 and Table 2, presumably "Total body dose" and "Integral total dose" are referring to the same thing - would stick to one term for consistency.
  6. Table 2c - why is total body dose not included in the modeling here, when it was included for 2a and 2b?

Reviewer 2 Report

This manuscript is well written, but rather technical analysis on RT dose to the immune cells and correlation to tumor control and OS in RTOG 0617. The topic is extremely important since the RT community is still puzzled about the RTOG 0617 results.

An important omission is the lack of information on the blood count: lymphopenia as well as leucopenia. Another important factor is the systemic treatment: chemotherapy and cetuximab regimen. Because the lymphopenia is influenced by the chemotherapy regimen administered and the cycles and intensity of the systemic treatment. Although the characteristic  ”received full chemo” was analyzed it should be presented in more detail. This should all be taken into the analysis. Currently it is mentioned in the discussion that “this study does not have blood count data”. However, the assumption that EDIC correlates with lymphopenia should be tested before any conclusion can be made.

The lymphocyte count and leucocyte count may be more predictive than the EDIC or ITDV.

There is room for more refinement. The EDIC calculation is based on MLD en MHD and an integral dose to all other irradiated tissues. The great bloodvessels (eg truncus pulmonalis) are containing a high concentration of lymphocytes and are contributing according to this EDIC calculation similar as ‘other’ irradiated tissues. This can be added to the remarks on the EDIC model in the discussion.

It is also puzzling that the EDIC  seems to have less impact on the PFS  than the LPFS, one might expect that if the hypothesis was correct, that also distant metastases would be increased if the EDIC  had an impact on immune response to  distant metastatic disease.

 Minor points:

 In table 1, for some parameters it is not clear in which direction  the comparison is: eg gender: male as HR=1, or female? Esophagitis grade 3:  higher degree HR=1 or low level of grade 3 HR=1?
